# Physically Encrypted Wireless Transmission Based on XOR between Two Data in Terahertz Beams

Hanwei Chen [1], Ming Che [1], Naoya Seiki [1], Takashi Shiramizu [1], Takuya Yano [1], Yuya Mikami [1], Yuta Ueda [2] and Kazutoshi Kato [1,*]

1 Graduate School of Information Science and Electrical Engineering, Kyushu University, Fukuoka 819-0395, Japan; chw_scut@yeah.net (H.C.); che.ming.677@m.kyushu-u.ac.jp (M.C.); seiki.naoya.703@s.kyushu-u.ac.jp (N.S.); yano.takuya.524@s.kyushu-u.ac.jp (T.Y.); mikami@ed.kyushu-u.ac.jp (Y.M.)
2 NTT Device Technology Laboratories, NTT Corporation, Atsugi 243-0198, Japan; yuta.ueda.dh@hco.ntt.co.jp
* Correspondence: kato@ed.kyushu-u.ac.jp; Tel.: +81-92-802-3753

**Abstract:** Future wireless communications require higher security as well as a higher data rate. We have been studying physically secured wireless transmission systems and previously proposed encryption/decryption techniques based on the AND operation caused by coherent detection between two encrypted data sequences on two different terahertz carriers. Furthermore, we suggested that by employing the XOR operation as the decryption, the proposed system can be made more secure because XOR increases the computational complexity for eavesdroppers to recover the plaintext. In this paper, we propose the XOR operation between two data sequences on FSK-modulated terahertz waves. By constructing the XOR encryption transmitters/receivers, which consisted of high-speed wavelength tunable lasers and arrayed uni-traveling-carrier photodiodes (UTC-PDs), we successfully demonstrated the XOR operation between two data sequences on 200 GHz waves from the two transmitters.

**Keywords:** coherent detection; phased arrays; photodiodes; photomixing; terahertz communications; wavelength tunable laser; wireless communication system security; XOR operation

## 1. Introduction

Recently, there has been growing interest in the research of terahertz (THz) waves as carriers in wireless communication. Compared with microwave and millimeter-wave frequency bands, the THz frequency band has more potential for capacity enhancement because of the large bandwidth. We have been researching THz wave generation by the photomixing of a two-tone lightwave and THz-wave communication based on this photonics technology. The uni-traveling-carrier photodiode (UTC-PD) utilizes only electrons as the active carriers. This unique feature is the key to its ability to achieve excellent high-speed and high-output characteristics simultaneously, making it one of the most promising photomixers for generating THz waves [1]. However, the limitation of radiation power at the UTC-PD caused by the space charge effect seems to pose a challenge to the communication system deployment [2–6]. As a way to overcome the limitation, arrayed UTC-PDs integrated with antennas were fabricated to increase the power as well as the directional gain. In our previous result, the coherently combined 300 GHz from eight-arrayed UTC-PDs revealed a peak power enhancement of 60-fold. Furthermore, a THz beam steering as wide as 50° was demonstrated by optical phase control [7]. These beam control technologies seem to have reached an applicable level for large-bandwidth wireless communication.

Information security in wireless communication is crucially important because the communication medium is always open to eavesdroppers. Normal cryptographic techniques operate at the upper layers of wireless networks higher than the medium access

control (MAC) layer and is commonly adopted by existing communication systems. However, it is pointed out that these encryption at the upper layers is not perfectly secure since the encrypted plaintext could be decrypted by an exhaustive key search [8]. Moreover, the advent of quantum computing threatens many commonly used cryptographic systems [9,10]. For example, a cryptographic system such as Rivest–Shamir–Adleman (RSA) can be broken by Shor's algorithms with enough quantum bits and quantum bit operations. Post-quantum cryptography (PQC), as well as its implementations for embedded systems, has received considerable scholarly attention to defend against quantum attacks [11,12]. To ensure the security of PQC systems, it is essential to consider both the algorithmic strength against quantum attacks and the resilience against side-channel attacks (SCAs) during the implementation phase. Specifically, FPGA-based systems can be prone to side-channel attacks due to their reprogrammability, making it challenging to protect against potential vulnerabilities. Similarly, ASICs, although designed for specific applications, can still be susceptible to side-channel attacks if appropriate countermeasures are not implemented.

On the other hand, physical layer security (PLS) safeguards information by exploiting the intrinsic characteristics of the communication mediums. Studies such as [13] have shown that the secrecy of a wireless communication system can be enhanced by PLS. Higher secrecy can be achieved by using the PLS technique compared with traditional cryptographic methods that are usually more computationally complex. PLS can also be plugged into existing security mechanisms to build more reliable wireless links. Furthermore, PLS techniques are suitable for high-frequency electromagnetic wave carriers that have good directivity of the emitted beam. The physical characteristics of a high-frequency electromagnetic wave suggest a higher resistance against eavesdropping. By employing hybrid beamforming in a multiple-input multiple-output (MIMO) millimeter-wave communication system, the security of the overall system can be improved [14]. There is also research on signals masked by the quantum noise ciphers. Although the transmission system is secured against eavesdropping because of the inevitable randomness of quantum noise measurement, it is vulnerable to known plaintext attacks [15]. Other researchers focus on increasing the directivity of electromagnetic waves. THz links created by a leaky wave antenna have been studied, showing potential for secured point-to-point wireless communication. However, it was pointed out that an eavesdropper can intercept signals by scattering the radiation toward the eavesdroppers [16]. The security of wireless links using vortex THz beams has been experimentally proved, but it is limited by the difficulty of generation and detection of the vortex beams in the THz spectral range [17].

Previously, we proposed a physically encrypted wireless transmission system based on beam steering and coherent detection in the THz band for a more secure connection against eavesdropping [18]. In the proposed system, data were encrypted into two data sequences that were carried by two THz beams and recovered by deducing an AND operation with each other at a coherent receiver. We furthermore suggested that XOR operation between two data sequences on the two THz beams would be more secure than AND operation in this system. Because the recovered bit of plaintext after AND operation is always "0" as long as either of the received bits of ciphertext is "0".The computational complexity is reduced for eavedroppers to recover the original plaintext. The XOR encryption/decryption scheme is also known as one-time pad encryption, which cannot be cracked under certain conditions [19]. By combining our proposed system with a one-time pad encryption scheme, illegal access by eavesdropping on the transmitting data can be avoided.

In this paper, a novel encrypting system based on XOR was proposed utilizing two data sequences modulated with frequency shift keying (FSK). Transmitters composed of four-channel arrayed UTC-PDs in a combination of wavelength tunable lasers were employed to generate FSK-modulated THz waves. With a mixer at the coherent receiver, the decryption based on XOR operation was demonstrated.

## 2. The Physically Encrypted Transmission

### 2.1. Configuration of the Proposed System

The proposed physically secured wireless transmission system consists of an encrypter, two radio-over-fiber feeding lines, two THz transmitters, and a decrypter, as shown in Figure 1. At the encrypter, the original data are randomly decomposed into two different data sequences (Data1 and Data2). For example, a randomly generated key sequence is used as Data2 while the XOR result between the key sequence and original data is Data1, so that the result of the XOR operation between Data1 and Data2 coincides with the plaintext M, which is regarded as a one-time pad (OTP) scheme. The OTP guarantees unconditional security as long as the key remains secret and is never reused. The ciphertext provides no information about the plaintext, making it highly secure. After the encryption, each data sequence is converted into a two-tone lightwave by electro-optic modulators. Here, the frequency spacing of the two-tone lightwave is in the THz band. At each transmitter (Tx1, Tx2), the two-tone lightwave is irradiated into the UTC-PD array to generate the same data sequence on a carrier frequency equal to the frequency spacing of the two-tone lightwave. From the antennas integrated with the arrayed UTC- PDs at Tx1 and Tx2, THz beams (THz1, THz2) are radiated to a target position. Due to the weaker effect of diffraction of THz waves, the beam width is comparatively narrow, which is made further directional by arrayed antennas. By phase manipulation of the THz waves, the beams can be steered to a certain angle. At the target position, the decrypter first generates the heterodyne signal between THz1 and THz2. Then, the envelope of the heterodyne signal is extracted so that Data1 and Data2 are decrypted into the original data sequence. The detailed decryption method is discussed in the following section.

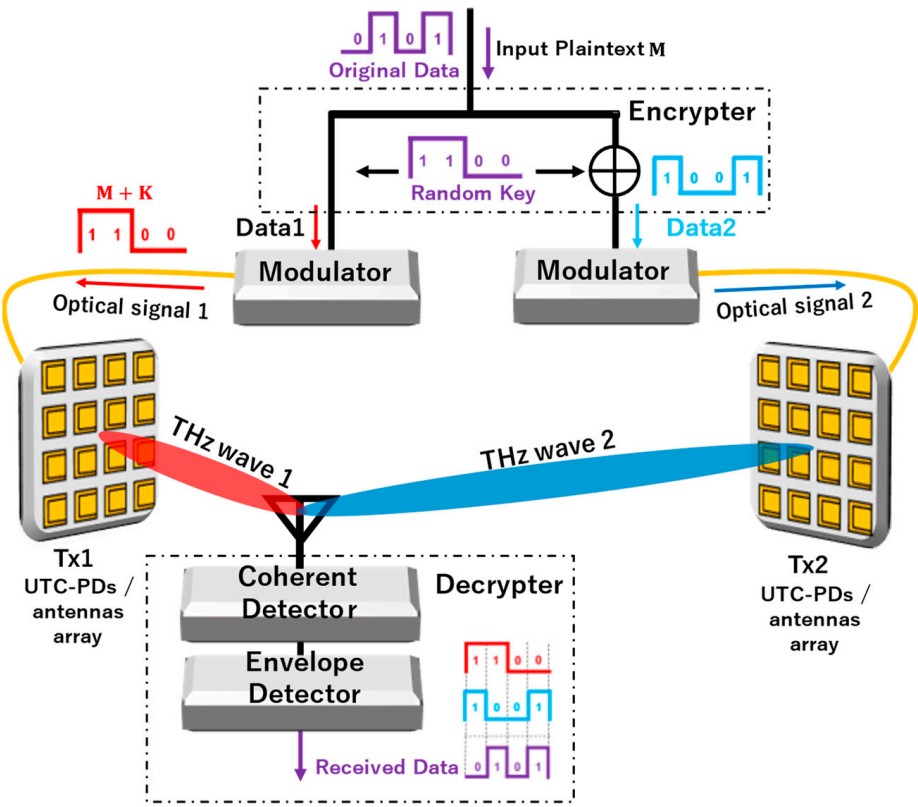

**Figure 1.** Conceptual diagram of the proposed high-security wireless transmission system.

To recover the original data correctly, both THz1 and THz2 need to be received simultaneously. In other words, correct data can only be decrypted within a limited area where both beams are overlapped. Since THz waves emitted from UTC-PD/antennas arrays are highly directional, the receivable area is significantly restricted. In addition, to

receive the data correctly, the receivers need to locate not only in the overlapped area of both beams but also at the position where phases of the two data sequences are matched with each other. Specifically, a phase mismatch of more than 1 bit can result in wrong decrypted data sequences. The position where the original data can be correctly detected is named the decodable zone in this work, as shown in the center of Figure 2a. Assume that the radiation power and the detected signal-to-noise ratio (SNR) are high enough for data transmission in the radiation direction. When we assume the 3 dB beam width as 7°, the distance between the two Txs as 100 m, and the radiation angle of both Txs as 45°, the area of the decodable zone vs. the bit rate is calculated as Figure 2b. As the bit rate increases, the decodable zone is further restricted as the tolerance of data phase mismatch between two data sequences reduces. For example, for a 10 Gbit/s signal, though both electromagnetic waves can be received by eavesdroppers within around 300 m², a phase-matched data pair can be detected only within less than 1 m². Therefore, eavesdropping becomes essentially impossible when the eavesdropper's antenna cannot be located within the decodable zone. Furthermore, for a higher-capacity link, the decodable zone could become smaller, making eavesdropping even harder.

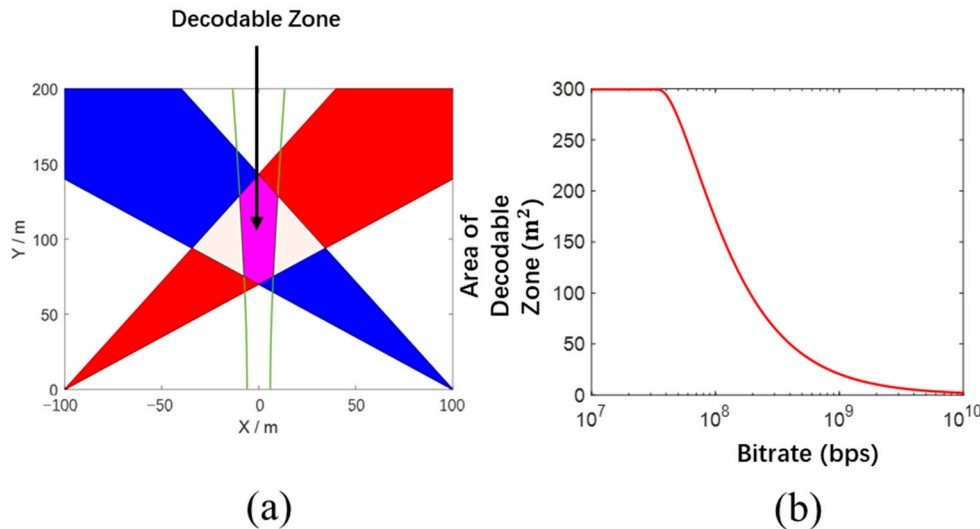

**Figure 2.** (**a**) Illustration of the decodable zone (pink color) and (**b**) the calculated area of the decodable zone vs. bit rate.

### 2.2. THz Beam Steering Enabled by Arrayed UTC-PD

Phased-array-based THz-wave beam steering is a potential technique for manipulating THz beams without mechanical scanning [20]. For THz wave generation by photomixing, the phase shift can be obtained before optical-to-THz conversion using mature and reliable fiber optic communication technologies. With the alignment of the optical phase offset of the lightwaves between adjacent channels in the phased array, a certain angle of beam steering can be achieved. Instead of controlling the phase of THz waves directly, the optical phased array controls the phase of lightwaves and thus manipulates the THz waves indirectly.

A phased-array-based beam steering system consists of a group of phase shifters before the antenna array. The relative phase of the signals sent to each antenna is carefully adjusted to achieve interference for the propagation of electromagnetic waves. An optical phased array (OPA) generates a gradient optical phase offset between lightwaves in different channels with these optical phase shifters. Following the introduction of each coupled lightwave onto the photomixer, the resulting THz wave exhibits the same frequency but a varying phase. Ultimately, the emission direction of the combined THz beam pattern is altered by the optical phase shift. Assuming the beam steering angle θ can be obtained

when the antennas are equally spaced with the distance *d*, the phase difference between adjacent channels $\Delta\phi$ can be expressed as

$$\Delta\phi = \frac{2\pi}{\lambda}dsin\theta,\tag{1}$$

where $\lambda$ is the wavelength of the radiated THz wave. In other words, controlling the wave interference of the array signals allows for the acquisition of higher directional gains in the angle of $\theta$.

We developed an InP-based monolithic chip consisting of an array of photomixers and antennas. Figure 3a shows the schematic of the antenna-integrated UTC-PD element. Each bowtie antenna consists of two triangular areas made of Au, as shown in Figure 3b, one of which is connected to the cathode of the UTC-PD, and the other is coupled to the anode. Figure 3c shows the cross-sectional view of an edge-coupled UTC-PD where the input light is refracted at an etched facet and introduced into the UTC-PD with an incident angle. Phased-array-based THz-wave beam steering is achievable by performing optical phase control before optical-to-THz conversion. The arrayed UTC-PDs pave the way for enhancing the directional gain and beam steering of THz waves, which is indispensable for high-speed and secured THz wireless communication.

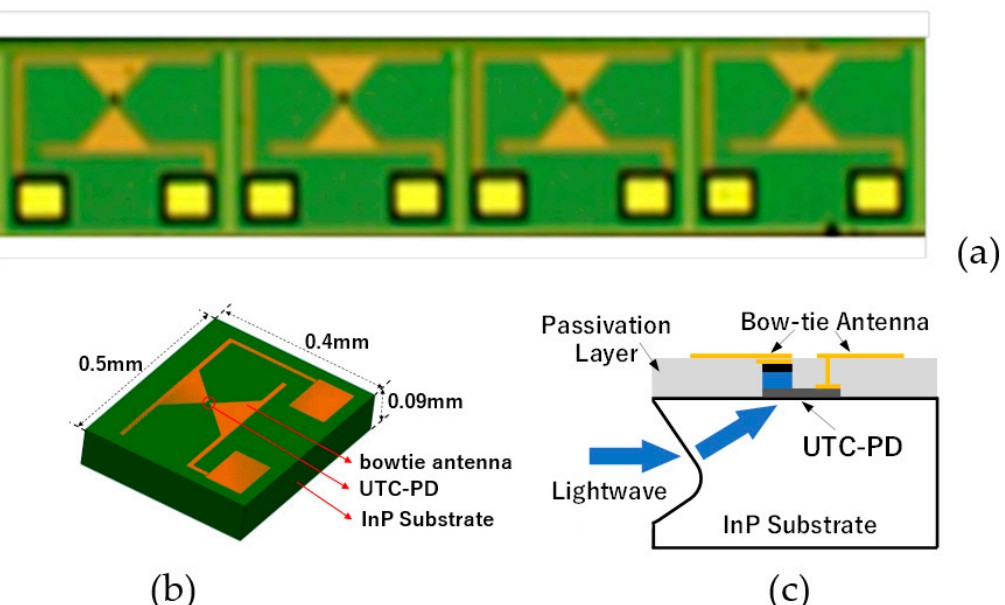

**Figure 3.** (**a**) Photo of arrayed UTC-PDs, (**b**) schematic of the UTC-PD integrated with bowtie antenna element, (**c**) cross-sectional schematic of the UTC-PD element.

### 2.3. Decryption Based on Coherent Detection

Suppose two spatially overlapped THz beams, THz1 and THz2, whose frequencies are $f_1$ and $f_2$, and phases are $\varphi_1$ and $\varphi_2$, respectively, are received by a coherent detector. The electric fields of these two THz waves are described as follows:

$$\begin{cases} E_{THz1}(t) = A_1 e^{-j(2\pi f_1 t + \varphi_1)} \\ E_{THz2}(t) = A_2 e^{-j(2\pi f_2 t + \varphi_2)} \end{cases},\tag{2}$$

where $A_1$ and $A_2$ are the amplitudes of THz1 and THz2. The detected power at the coherent detector can be expressed as

$$P_{det} \propto |E_{THz1} + E_{THz2}|^2 \sim \frac{A_1^2}{2} + \frac{A_2^2}{2} + A_1 A_2 \cos\{2\pi(f_1 - f_2)t + (\varphi_1 - \varphi_2)\}.\tag{3}$$

Thus, the intermediate frequency (IF) $f_1 - f_2$ is generated in the case that $f_1 \neq f_2$, which is referred to as heterodyne detection. Figure 4 shows the schematic of the result of coherent detection between THz1 and THz2 when they are frequency-modulated with different data sequences, 0101 and 0011, respectively. Here, the carrier frequencies $f_H$ and $f_L$ correspond to binary data "1" and "0", respectively. When a coherent detector receives both THz waves with different frequencies, the IF with a frequency of $f_{IF} = f_H - f_L$ is generated as described above. Meanwhile, when their frequencies are almost the same, $f_{IF}$ is out of the IF band. Thus, almost no $f_{IF}$ can be detected. In such a configuration, the waveform representing the result of the XOR operation between two ciphertexts can be obtained by converting the $f_{IF}$ into baseband using a envelop detector. As a result, two ciphertexts are decrypted by bit-wise XOR using our proposed system. Both down-conversion and decryption are completed in the physical layer, providing high compatibility with higher-layer cryptographic algorithms and a simple configuration of receivers suitable for mobile communication.

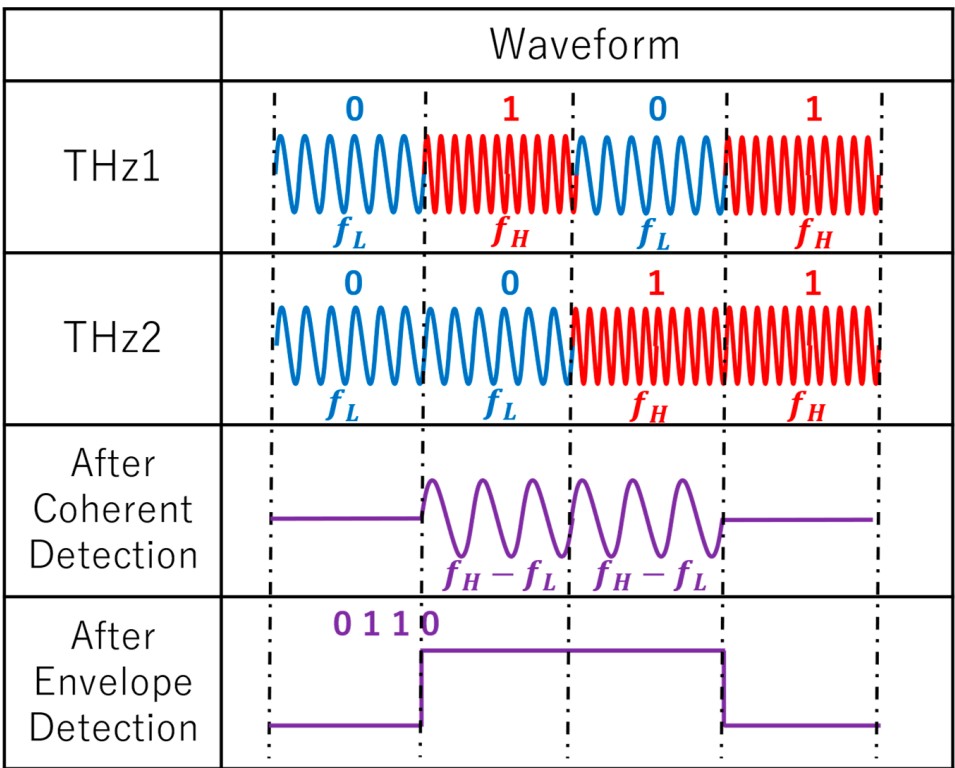

**Figure 4.** XOR operation enabled by coherent detection.

### 2.4. XOR Encryption Compared with AND Encryption

It is pointed out that for our previously proposed transmission configuration, when one of the transmitting signals is eavesdropped on, parts of the plaintext can be recovered by the eavesdroppers. This is because the plaintext is recovered by deducing bitwise AND operation between two received ciphertext. Whenever the bit in the ciphertext is "0", the corresponding bit in the plaintext is also "0", making it insecure when one of the channels is eavesdropped on. On the contrary, the XOR operation guarantees the security even if one of the channels is eavesdropped on. Therefore, we proposed a new secure transmission system based on the XOR operation. Two channels of frequency-modulated signals are emitted from transmitters and overlap at target receivers which are composed of a coherent detector and envelop detector by which the ciphertext is received and decrypted into plaintext at the same time. Compared with the configuration with AND encryption, our new proposed system eliminates possible unauthorized access by eavesdropping on one of the channels while maintaining the simple configuration of the receivers.

## 3. Experimental Setup

Figure 5 shows the schematic configuration of a feasibility demonstration of the XOR operation. First, a lightwave was modulated with FSK to make a data sequence by using an electro-optically tunable reflection-type transversal filter (RTF) laser that can switch the lasing wavelength within 500 ps [21,22] using the fast tuning response of the electro-optic effect. The pattern data from the pulse pattern generator (PPG) with 200 Mbit/s were amplified by a power amplifier and applied to the wavelength-tuning electrode of the RTF laser, in which the lasing wavelength was changed by the applied voltage. For utilizing an IF of 4.2 GHz at the receiver end, the applied voltage was set at 3.0 V (low) and 5.5 V (high). Another laser diode was employed to lase a wavelength for generating the beat signal with a frequency of 200 GHz and 204.2 GHz so that the frequency of generated THz waves by photomixing was switched between 200 GHz and 204.2 GHz to be an FSK data sequence.

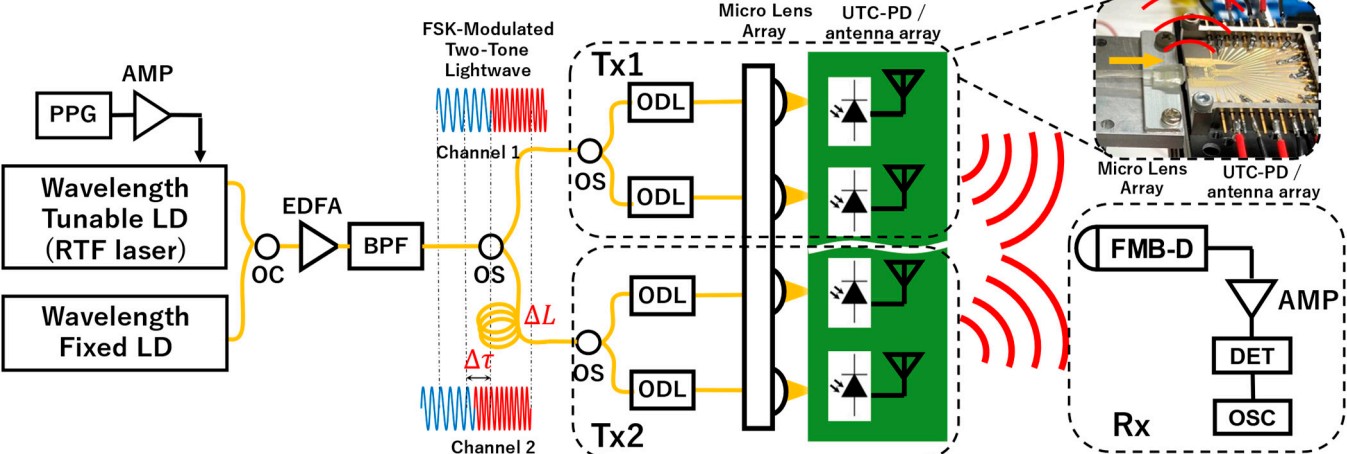

**Figure 5.** Experimental setup to verify the XOR operation.

Second, the lightwave from the two lasers was coupled by an optical coupler (OC). To overcome the low optical conversion efficiency of UTC-PDs, an erbium-doped optical fiber amplifier (EDFA) was employed to maximize the power enhancement of the two lightwaves with different frequencies. Meanwhile, a band-pass filter (BPF) was used for the removal of amplified spontaneous emission (ASE). Afterward, the signal was split into two channels (channel 1 and channel 2) by an optical splitter (OS). On one of the channels, an extra length $\Delta L$ of fiber was added so that the data sequences with data timing difference of $\Delta\tau$ were created at the transmitters. Then, the signal for each transmitter was further split into two paths, and every optical path had an optical delay line (ODL) that tuned the phase of the terahertz wave. The influence of the lightwave's polarization on the power of the generated THz wave and the coupling effect of THz waves was evident. As a result, polarization-maintaining fibers (PMFs) were employed to establish connections between different experimental equipment. After the ODL, each pair of lightwaves was introduced into a fiber-pigtailed microlens array and finally into the UTC-PD. At each transmitter, THz waves with the data sequences were emitted from the bowtie antennas integrated with the arrayed UTC-PD chip and combined in the direction toward the receiver by tuning the THz-wave phase with the ODLs.

At the receiver end, the two data sequences on two THz waves from both channels were received by a Fermi-level managed barrier diode (FMB-D) packaged with a broadband trans-impedance amplifier as a coherent detector [23]. To minimize the interference caused by standing waves, a thin broadband-type radio absorber operating at 300 GHz was utilized. It was attached to the outer curved surface of the FMB-D module, leaving the aperture unaffected. The FMB-D not only down-converted the THz waves into the IF signal, but also acted as a decrypter, revealing the plaintext. Then, an envelope detector (DET) converted

the IF signal into baseband signal that equal to the XOR between two received ciphertext. Finally, the waveforms were sampled and observed by an oscilloscope (OSC).

## 4. Result and Future Directions

### 4.1. Experiment Result and Discussion

The 200 GHz/204.2 GHz FSK data sequence with an alternating pattern at a bit rate of 200 Mbit/s was generated by the PPG, the RTF laser, and the UTC-PD, with a transmission distance of 15 cm. The output power of each THz wave was about 10 μW. THz waves, whose power was less than 1 μW, were detected in the FMB-D. The extra fiber length of channel 2 $\Delta L$ was set as follows:

$$\Delta L = \frac{c\Delta\tau}{n}, \tag{4}$$

where $c$ is the speed of light and $n$ is the effective refractive index of the optical fiber. $\Delta\tau$ refers to the delay time between the signals of channel 1 and channel 2. Therefore, when we set $\Delta\tau$ as $\Delta\tau$ =1.7 ns and $\Delta\tau$ =2.5 ns, which is 1/3 bit and 1/2 bit for 200 MHz NRZ signals, the extra fiber length $\Delta L$ should be 33 cm and 50 cm. The first and second rows of Figure 6 illustrate the modulation data at each channel. The detected IF signals after coherent detection and subsequent envelope detection are shown in the third and fourth rows, respectively. At the waveforms after coherent detection, IF signals with a frequency of 4.2 GHz were observed only when the frequency of both channels was different, corresponding to our expectations. On the other hand, when the frequency of both channels was the same, IF signals of several megahertz were also detected, which was thought to be caused by the spectral linewidth of the RTF laser. This noise can be removed by introducing a proper band-pass filter or trans-impedance amplifier in the FMB-D. At the waveforms after envelope detection, NRZ waveforms with a frequency of 200 Mbit/s were observed. The duty ratio was 1/3 when the time delay $\Delta\tau$ was set to 1.7 ns and tended to be 1/2 when the $\Delta\tau$ was changed to 2.5 ns. Bits were "1" only when transmitting bits (or frequency) from both channels were different from each other, as explained above. Thus, the XOR operation was successfully deduced between two data sequences. These results demonstrated the principle of our proposed XOR decryption system.

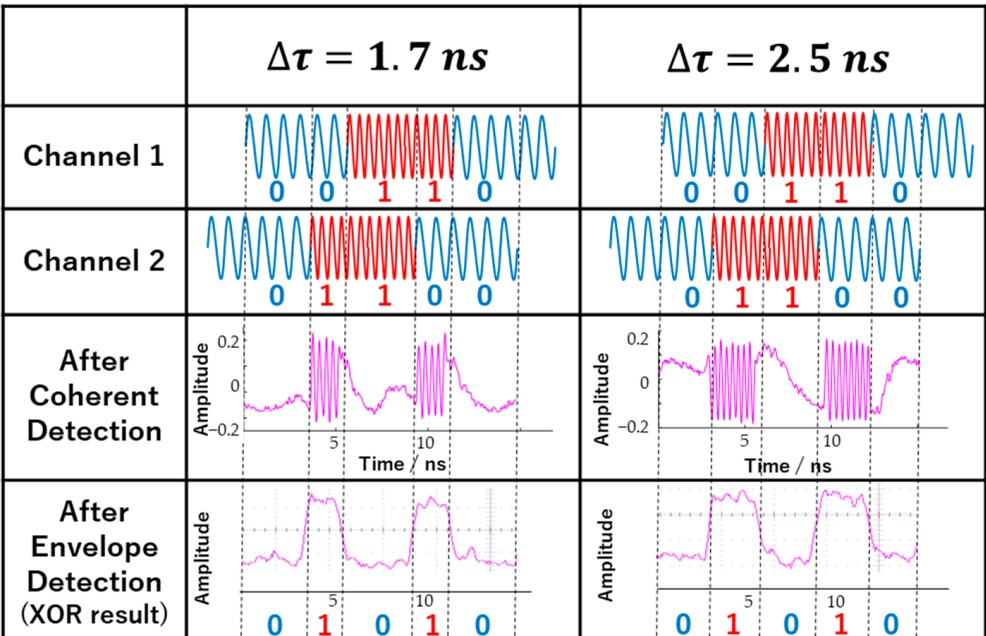

**Figure 6.** Illustration of the electric field of channel 1 (1st row) and channel 2 (2nd row), detected IF signals after coherent detection (3rd row), and resulting waveforms after envelope detection (4th row).

*4.2. Future Directions*

Embedding our proposed system with other cryptographic systems is a possible choice for the real-world application. The advantages of our proposed system are obvious against most of the passive eavesdropping attacks because the received signal power at the legitime user is far above that of the eavesdroppers. However, when encountering active eavesdropping, such as pilot contamination attacks, our system may wrongly steer the beams towards the eavesdroppers. Therefore, countermeasures against active eavesdropping need to be taken. For example, detection schemes such as the cooperative detection scheme used in MIMO systems described in [24] can be applied to our proposed system. Except for eavesdropping, strategies to fight against other passive/active attacks such as jamming and SCAs are also crucial [25]. Specifically, SCAs as fault attacks or power analysis attacks can be tackled by adding dummy operations, using specific parameterizations or fault detection schemes, as described in [26,27]. Fortunately, owing to the high adaptability of our proposed system, it can be combined with those countermeasures that are taken in the upper layer of the network.

The development of devices with a higher operating frequency is needed to demonstrate a higher bit rate transmitting capability. First, for the results obtained in this work, the bit rates were limited by the envelope detectors at the receiver end in our setup. As a possible solution, we are trying to design envelope detectors with an operating frequency of over 60 GHz. Secondly, as the bit rates increase, the required output power escalates as well. Due to the saturation effect and thermal effect of a single UTC-PD, it is worth considering increasing the elements of the UTC-PDs array, which will increase the directivity of the emitting beams at the same time. Finally, the driving devices of the RTF laser need to be redesigned to adapt to a higher operating frequency.

Finally, the functionality of our proposed system as a secure communication system needs further demonstration in terms of link budget and secrecy capacity.

## 5. Conclusions

A novel physically secured transmission system was proposed based on XOR operation between two data sequences on two THz beams. By making use of the highly directional THz waves and OTP scheme, eavesdropping attacks on the transmission system can be theoretically prevented. Compared with the previously proposed system using AND operation, the possibility of unauthorized access by eavesdropping on one of the channels can be eliminated, thus enhancing the security against passive eavesdroppers without increasing complexity at the receiver end. Experiments were conducted to demonstrate the feasibility of the XOR operation as decryption using binary FSK-modulated THz waves generated by an electro-optically tunable RTF laser and UTC-PD array integrated with bowtie antennas. The frequency difference between two discrete frequencies was set as 4.2 GHz with both frequencies in the 200 GHz band. A 200 Mbit/s NRZ data sequence was modulated in the two-tone lightwaves and divided into two channels between which a certain time delay was added. Pairs of lightwaves were converted into THz waves by two arrayed UTC-PDs with an average output power of 10 μW and spatial overlap at the target receiver located 15 cm away from the transmitters. The data sequences after XOR operation were represented by the duty ratio of the decrypted signals. The decrypted signals corresponded well to the theoretical bit-wise XOR of two transmitting data sequences, showing that the XOR decryption can be correctly performed by our proposed system. The objective of our future work is to further demonstrate the feasibility of the proposed system with a higher bit rate.

**Author Contributions:** Conceptualization, H.C. and K.K.; methodology, H.C., M.C. and K.K; validation, H.C., M.C., N.S., T.S., T.Y. and K.K.; investigation, H.C. and K.K.; resources, Y.U. and K.K.; data curation, H.C.; writing—original draft preparation, H.C.; writing—review and editing, Y.M. and K.K.; visualization, H.C.; supervision, Y.M. and K.K.; project administration, K.K.; funding acquisition, K.K. All authors have read and agreed to the published version of the manuscript.

**Funding:** This work was supported in part by the commissioned research by the MIC/SCOPE #195010002, National Institute of Information and Communications Technology (NICT) #02801, #00901, and JSPS KAKENHI grant number JP20H00253, JP21K18730.

**Data Availability Statement:** Data are available on request.

**Conflicts of Interest:** The authors declare no conflict of interest.

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
