# Peer review of "Physically Encrypted Wireless Transmission Based on XOR between Two Data in Terahertz Beams"

_electronics, doi:10.3390/electronics12122629_

Round 1
Reviewer 1 Report
Some concerns of the proposed method are suggested to be addressed:
(1) Please provide the explanation on the term "UTC-PD".
(2) The format and grammar of the article are suggested to be modified.
(3) It is suggested to add more details on the circuit design.
(4) The link budget analysis of the transceiver is suggested to be added.
(5) A comparison table comparing the proposed technique with other state-of-the-art designs is suggested to be added.
Author Response
Response to the comments about the submitted paper electronics-2429717
Title: Physically Encrypted Wireless Transmission Based on XOR Between Two Data in Terahertz Beams
Authors; Hanwei Chen, Ming Che, Naoya Seiki, Takashi Shiramizu, Takuya Yano, Yuya Mikami, Yuta Ueda, Kazutoshi Kato
To Editor and reviewers of MDPI electronics
Dear editor and reviewers,
We thank the reviewer for his or her constructive comments. We have addressed all of them and modified the paper accordingly using the “Track Changes” function. Please find attached our revised manuscript. Our point-by-point responses to the reviewer’s comments are given below.
Please note that the reviewer’s comments are green while our answer are not. The revisions to the original manuscript are indicated in blue. At the revised manuscript, the sentences which are modified or added are marked with a yellow marker.
Best regards,
Chen Hanwei

Reviewer 2 Report
Why the research in the paper is important:
Future wireless communications require higher security as well as higher data rate. Since wireless communication is vulnerable to eavesdroppers due to its transmission in an open area, information security is one of the utmost important issues for actual applications. Among the techniques of safeguarding information security, physical layer security (PLS) is a promising one to improve the reliability of point-to-point links for next-generation wireless networks.
Contributions and novelty of the paper with respect to the state of the art:
The authors have been studying physically secured wireless transmission systems and previously proposed encryption/decryption techniques based on AND operation caused by coherent detection between two encrypted data sequences on two different terahertz carriers. Besides, they suggested that by employing XOR operation as the decryption, the proposed system can be made more secured. Because XOR increases the computational complexity for eavesdroppers to recover the plaintext. In this paper, they proposed XOR operation between two data sequences on FSK-modulated terahertz waves. By constructing the XOR encryption transmitters/receivers, which consisted of high-speed wavelength tunable lasers and arrayed uni-traveling-carrier photodiodes (UTC-PDs), they successfully demonstrated the XOR operation between two data sequences on 200 GHz waves from the two transmitters.
The comments with respect to shortcomings and to improve the paper quality of the state-of-the-art:
The followings are the Major issues might include problems with the study’s methodology, techniques, analyses, missing controls or other serious flaws. Please address them carefully to avoid multiple revisions.
- With any new security measure implementation, you need to make sure you provide benchmark for active/passive side-channel attacks (SCAs). Fault attack and power analysis attack and countermeasures need to be mentioned. Moreover, combined attacks need to be mentioned. I would like to see a paragraph on combined fault and power analysis attacks assessment and countermeasure. At least add a paragraph to describe it, that is enough. Mention and add works about "fault detection of ring-LWE on FPGA" too.
- With the advent of post-quantum cryptography (PQC), it is better to add some relevant works to make sure you cover that topic too. This is the hottest topic in cryptography now. When PQC replaces ECC/RSA every security application from smart phones to block chains will be affected. With PQC, ARM Cortex M4 and Cortex-A implementations are important for embedded systems, add previous work on: Curve448 and Ed448 on Cortex-M4, SIKE on Cortex-M4, SIKE Round 3 on ARM Cortex-M4, Kyber on 64-Bit ARM Cortex-A.
- Please add comparisons in a table (or subsection) so that one could fairly compare your work with similar previous works
- NIST lightweight standardization was finalized in Feb. 2023. Also mention fault attacks as side-channel attacks, these topics to explore and add references. Fault detection of architectures of Pomaranch cipher, reliable architectures of grostl hash, fault diagnosis of low-energy Midori cipher, fault diagnosis of RECTANGLE cipher.
- References are not uniformly formatted.
- Please add a subsection and one or more future works for enhancing your presentation
- DPA+DFA for example can be mounted at the same time and their combined countermeasures (for example TI and Error Detection Schemes) can be used for thwarting attacks in combined manner (need to be discussed). Adding a paragraph suffices.
N/A
Author Response

(The authors gave the same response as above.)

Round 2
Reviewer 1 Report
All my concerns have been addressed in this version.
Reviewer 2 Report
Authors have addressed some of my comments but they need to address all the followings as well.
This is an acceptable paper on important topic, so you need to include prior works references for each of these separately, add a recent reference/work on PQC and on fault detection of each of these separately: (a) Kyber on 64-Bit ARM Cortex-A, (b) Cryptographic accelerators on Ed25519.
Same thing for fault attack of lightweight ciphers, add for each of these a reference: (a) Fault detection of architectures of Pomaranch cipher, (b) reliable architectures of grostl hash, (c) fault diagnosis of RECTANGLE cipher.
Can you also comment on hardware/software platforms, ASIC vs. FPGA vs. ARM/RISC-V, adding 1-2 sentences is enough.
Round 3
Reviewer 2 Report
My comments are addressed.